# Saposin C, Key Regulator in the Alpha-Synuclein Degradation Mediated by Lysosome

**DOI:** 10.3390/ijms231912004

**Published:** 2022-10-09

**Authors:** Clara Ruz, Francisco J. Barrero, Javier Pelegrina, Sara Bandrés-Ciga, Francisco Vives, Raquel Duran

**Affiliations:** 1Department of Physiology and Institute of Neurosciences “Federico Olóriz”, Centre of Biomedical Research, University of Granada, 18016 Granada, Spain; 2Instituto de Investigación Biosanitaria ibs.GRANADA, 18012 Granada, Spain; 3Movement Disorders Unit, University Hospital Clinic San Cecilio, 18016 Granada, Spain; 4Molecular Genetics Section, Laboratory of Neurogenetics, National Institute on Aging, National Institutes of Health, Bethesda, MD 20898, USA

**Keywords:** Parkinson’s disease, lysosomal dysfunction, β-glucocerebrosidase, PSAP, Cathepsin D

## Abstract

Lysosomal dysfunction has been proposed as one of the most important pathogenic molecular mechanisms in Parkinson disease (PD). The most significant evidence lies in the *GBA* gene, which encodes for the lysosomal enzyme β-glucocerebrosidase (β-GCase), considered the main genetic risk factor for sporadic PD. The loss of β-GCase activity results in the formation of α-synuclein deposits. The present study was aimed to determine the activity of the main lysosomal enzymes and the cofactors Prosaposin (PSAP) and Saposin C in PD and healthy controls, and their contribution to α-synuclein (α-Syn) aggregation. 42 PD patients and 37 age-matched healthy controls were included in the study. We first analyzed the β-GCase, β-galactosidase (β-gal), β-hexosaminidase (Hex B) and Cathepsin D (CatD) activities in white blood cells. We also measured the GBA, β-GAL, β-HEX, CTSD, PSAP, Saposin C and α-Syn protein levels by Western-blot. We found a 20% reduced β-GCase and β-gal activities in PD patients compared to controls. PSAP and Saposin C protein levels were significantly lower in PD patients and correlated with increased levels of α-synuclein. CatD, in contrast, showed significantly increased activity and protein levels in PD patients compared to controls. Increased CTSD protein levels in PD patients correlated, intriguingly, with a higher concentration of α-Syn. Our findings suggest that lysosomal dysfunction in sporadic PD is due, at least in part, to an alteration in Saposin C derived from reduced PSAP levels. That would lead to a significant decrease in the β-GCase activity, resulting in the accumulation of α-syn. The accumulation of monohexosylceramides might act in favor of CTSD activation and, therefore, increase its enzymatic activity. The evaluation of lysosomal activity in the peripheral blood of patients is expected to be a promising approach to investigate pathological mechanisms and novel therapies aimed to restore the lysosomal function in sporadic PD.

## 1. Introduction

Parkinson’s disease (PD) is the second most common neurodegenerative disease and the first in terms of movement disorders. PD affects about 1–2 individuals out of 1000 in the population at any age and one percent in the population over 60 years, with age being the main risk factor for its development [1]. Despite its high prevalence, there are no specific biomarkers for early diagnosis to date, which remains primarily clinical. Motor and non-motor clinical symptomatology is mainly due to a marked loss of dopaminergic neurons of the Substantia nigra pars compacta (SNpc) and to the formation of cytoplasmic aggregates of α-Synuclein protein (α- Syn), named Lewy bodies [2]. About 10% of all PD cases have been linked to predisposing genes that can be inherited either in an autosomal dominant or recessive manner; however, the vast majority of cases, about 90%, are sporadic, the cause of which still remain unclear [3]. Many of the PD-related genes in the familiar form code for proteins involved in proteasomal and lysosomal degradation pathways, including PINK-1, Parkin (ubiquitin ligase), UCHL1 (ubiquitin hydrolase L1), DJ-1 (chaperone activity), and LRRK2 (kinase activity) [4,5,6]. Therefore, lysosomal dysfunction has been recently involved in the pathogenesis of PD and other synucleopathies [7]. One of the most significant gene associated with PD is *GBA*, which encodes for the lysosomal enzyme β-glucocerebrosidase (β-GCase).

Heterozygous mutations in the *GBA* gene have been considered as the main genetic risk factor that predisposes to the development of PD [8]. The discovery of this association took place in the clinic, with the identification of parkinsonian symptoms in patients with Gaucher disease (GD). GD is a lysosomal storage disorder caused by biallelic mutations in the same *GBA* gene. Further studies also determined that in relatives of these patients (many of them being carriers of *GBA* mutations), there was a higher incidence of PD [9]. Cellular and animal models with deficiency in β-GCase activity tend to accumulate α-Syn in the cytoplasm. It has been suggested that this enzyme binds directly to α-Syn in the lysosome, and contributes to the stabilization of oligomeric soluble forms, considered highly neurotoxic [10]. On the other hand, other studies have shown that α-Syn forms a bond to the lysosomal membrane and interacts with β-GCase, resulting in its inhibition [11]. Gegg et al. [12] reported a reduction in the enzymatic activity of β-GCase in the brain of PD patients with mutations in *GBA*, as well as in sporadic cases. Subsequently, Murphy et al. [13] corroborated these results, demonstrating that there was a deficit of β-GCase activity in the brain of patients with sporadic PD, being responsible for the accumulation of α-Syn and alterations in lipid metabolism. In a recent work, Bae et al. [14] demonstrated that β-GCase depletion promotes the propagation of α-Syn aggregates between cells, suggesting that this toxic role could be liable for the disease progression. All these results clearly indicate a strong relationship between β-GCase activity and α-Syn, although it remains to be determined whether the formation of protein aggregates is consequence of a failure in the lysosomal activity or, conversely, that the increase in α-Syn concentration is promoting the β-GCase impairment. However, this mechanism seems to be reversed in presence of Saposin C [15]. Saposin C is a non-enzymatic activator protein derived from Prosaposin (PSAP). Saposins act by facilitating the interaction between lysosomal enzymes, causing those sphingolipids to be degraded [16,17]. In particular, Saposin C plays an activator role in the hydrolysis of glucosylceramide by β-GCase and galactocerebroside by β-galactosidase (β-gal) [18]. These PSAP-derived activator proteins have been associated with different lysosomal storage disorders [19].

Here we aimed to study the contribution of the lysosome to the pathogenesis of PD through the analysis of a group of lysosomal enzymes (β-glucocerebrosidase (β-GCase; EC 3.2.1.45), β-galactosidase (β-gal; EC 3.2.1.23), β-hexosaminidase (Hex B; EC 3.2.1.52), and Cathepsin D (CatD; EC 3.4.23.5)) in peripheral blood from sporadic PD patients in the early stages of the disease. Furthermore, we measured PSAP and Saposin C activator protein levels. Thus, we expect to elucidate how the lysosomal dysfunction may contribute to α-Syn aggregation.

## 2. Results

Demographic and clinical data are summarized in the Table 1. The mean age at PD onset was 60.06 ± 10.77 years (mean ± SEM). All PD patients recruited were categorized as early I and II disease stages according to the Hoehn and Yahr scale. Total WBC protein concentration was significantly higher in PD patients compared to controls. There were no statistical differences between groups for gender and age.

### 2.1. Evaluation of the Lysosomal Enzymatic Activities

The β-GCase, β-gal, Hex B and CatD enzymatic activities in WBC from controls and PD patients are reported in Figure 1. 

β-GCase and β-gal enzymatic activities were significantly lower in PD patients compared to controls (β-GCase: controls = 13.02 ± 0.89 nmol/min/mg; PD = 10.44 ± 0.63 nmol/min/mg; *p* < 0.05; β-gal: controls= 20.37 ± 1.06 nmol/min/mg; PD = 18.02 ± 0.10 nmol/min/mg; *p* < 0.05), both showing a reduction of 20%. (Figure 1 A,C). No significant differences were found for Hex B activity between PD patients and controls (controls= 0.98 ± 0.09 U/mL; PD = 0.90 ± 0.12 U/mL; *p* = 0.122) (Figure 1B). Surprisingly, CatD activity was significantly higher in PD patients than in controls (controls = 51,646 ± 11,320 UFR/mg/mL; PD = 102,024 ± 11,985 UFR/mg/mL, *p* < 0.001), reaching an increase of 100% (Figure 1D). 

We also examined whether age and gender were associated with lysosomal enzymatic activities among PD patients and controls, but we did not find any significant association. 

### 2.2. Determination of GBA, BGAL, HEXB, CTSD, and α-Syn Protein Levels

We determined the protein levels of the above referred enzymes through Western blot analyses (Figure 2). We also evaluated PSAP and Saposin C protein levels, since they are the natural cofactors of both β-GCase and β-gal enzymes, and α-Syn protein level, whose metabolism is linked to lysosome activity (Figure 3). 

Blots revealed no significant differences in GBA and HEXB protein levels between PD patients and controls (Figure 2B,C). Conversely, BGAL and CTSD protein levels were significantly higher in PD patients, with an increase of 50% compared to controls (Figure 2C,D). We also explored BGAL and CTSD protein levels within PD patients, distinguishing between HY stages I and II, and observed that only PD patients at HY stage I showed a significant increase respect to controls.

In relation to PSAP and Saposin C protein levels, both experienced a significant decrease of approximately 40% in PD patients compared to controls (PSAP: controls = 1.139 ± 0.2170; PD = 0.6878 ± 0.05014, *p* < 0.05; Saposin C: controls = 0.2255 ± 0.04037; PD = 0.1203 ± 0.01337, *p* < 0.05). No differences were found between disease stages (Figure 3C,D).

Finally, as expected, a higher level of α-Syn protein was found in PD patients in comparison to controls (controls = 0.7898 ± 0.1195; PD= 1.477 ± 0.1738, *p* < 0.01), which was consistent with the disease progression (Figure 3E).

In order to explore possible associations between the assessed proteins, we performed Spearman’s Rank Correlation Coefficients. The results showed a strong negative correlation between PSAP and α-Syn protein levels (r = −0.6357, *p* = 0.01), in addition to a significant positive correlation between CTSD and α-Syn protein levels (r = 0.6485, *p* = 0.04) (Figure 4A,B).

## 3. Discussion

A number of recent investigations have reported a close relationship between lysosomal dysfunction, α-Syn aggregation, and PD [20,21]. In this sense, lysosome plays an essential role in α-Syn degradation, and furthermore, within the lysosomal enzymes, β-GCase and CatD have been directly involved in the α-Syn metabolism. Reduced β-GCase activity has been observed in the brain and cerebrospinal fluid of PD patients with GBA mutations, as well as in sporadic PD patients [12,13,22]. However, when this activity is explored in peripheral tissues, results are incongruous [23,24]. In this study, we provide evidence supporting that PD is associated with impaired lysosomal enzymatic activity, along with increased α-Syn and reduced PSAP and saposin C protein levels. We observed a significant decrease in both β-GCase and β-gal enzymatic activities in the WBC lysates of PD patients, whereas CatD enzymatic activity was significantly increased in comparison to controls. No significant difference was found for Hex B enzymatic activity. Moreover, when we explored their corresponding protein levels, significant changes were only observed for BGAL and CTSD proteins, as they were increased in the WBC lysates of PD patients in comparison to controls. Supporting our results, as is mentioned above, reduced β-GCase activity has been reported in PD patients compared to controls [22,25,26,27]. This decreased activity seems to be promoting the accumulation of α-Syn, compromising the protein degradation capacity of lysosome and consequently favoring the formation of α-Syn aggregates. The cause underlying the decrease in β-GCase activity might be due to an impaired trafficking of GBA through the endoplasmic reticulum (ER)/Golgi apparatus (GA) that hinders its correct localization in the lysosome [12,28]. Although the aim of this study was not to investigate how GBA and α-Syn interact, we reported a significant increase of α-Syn levels in PD patients in comparison to controls. Likewise, β-gal activity was significantly reduced in the WBC lysates of PD patients compared to controls. Previous studies have reported similar results in the substantia nigra [29] as well as in CSF [26,27] of PD patients. Furthermore, the decrease observed in both β-GCase and β-gal activities was linked with a significant reduction in the PSAP and Saposin C protein levels. Surprisingly, we also found a negative correlation between PSAP and α-Syn protein levels. Taken together, the depletion observed in these enzymes might result from a reduction in PSAP and Saposin C protein levels. Since Saponsin C deficiency has been associated with GD and GD-like diseases [30,31,32], several studies have investigated the contribution of this natural cofactor to PD. Although there is no strong evidence that supports a genetic association between rare or common variants in *PSAP* gene and PD [33,34], several cell models have showed how Saposin C protects β-GCase from protease degradation in the lysosome and prevents its inhibition mediated by α-Syn [35,36]. In a recent work on peripheral blood mononuclear cells of PD patients, Avenali et al. [37] found a significant reduction in Saposin C levels in PD patients carrying *GBA* mutations in comparison to non-mutated PD patients but not versus healthy controls. However, exosomal α-Syn protein levels were significantly elevated in both *GBA*-PD and non-mutated PD patients compared to healthy controls. The deficiency observed in Saposin C could be a consequence of a negative feedback loop from increased α-Syn protein levels. It has been shown that α-Syn perturbs protein trafficking machinery downstream at the GA, resulting in the accumulation of immature proteins [28,38]. Thus, it is possible that PSAP does not correctly reach the lysosome where it is processed into active saposins, contributing in part to the reduction in β-GCase activity observed in PD patients. 

The lysosomal enzyme CatD has recently emerged as a key mediator in α-Syn proteolysis [39,40]. Huarcaya et al. [41] showed that the amount of mature CTSD protein was proportional to the reduction of α-Syn protein level in H4 neuroglioma cells overexpressing α-Syn protein. Moreover, Chu et al. [42] found a higher reduction in CTSD expression levels of nigral neurons positive for α-Syn inclusions in PD patients compared to healthy subjects. Here, we found that: (1) CatD enzymatic activity was significantly increased in PD patients compared to controls, being also parallel to its protein levels, and (2) there was a significant positive correlation between both CTSD and α-Syn proteins. These results were surprisingly opposed to the most scientific evidence focused on lysosomal enzymes, in which CatD activity and CTSD protein levels were reduced in PD [26,27,42]. Nevertheless, when we explored CTSD expression taking PD progression into consideration, it was notably higher in PD patients at HY stage I, being similar to controls in PD patients at HY stage II. It is possible that this observed increase responds to a positive regulation mechanism in an early attempt of the lysosome to restore the α-Syn clearance, and subsequently, in advanced stages, CTSD shows a reduction [43]. Several studies have documented that endogenous ceramide levels in endo-lysosomal compartments regulate CatD enzymatic activity. Ceramides seem to interact with pre-pro CTSD isoforms, promoting their proteolytic processing to mature and active isoforms [44,45]. Thus, the expected increase in monohexosylceramides derived from the deficiency of both β-GCase and β-gal activities observed in PD patients, might induce a rapid activation of CTSD, eventually resulting in the higher activity of the enzyme. 

In conclusion, our study brings to light the contribution of the lysosomal dysfunction to the pathogenesis of sporadic PD, showing a pattern of enzymatic activity that could be specific to early stages of the disease. It is remarkable that the significant decrease in Saposin C protein levels is affecting directly GBA function and α-Syn clearance. Although we are aware that these results might not be completely reproducible in the brain, and they also need to be replicated in larger independent cohorts, including early and advanced PD patients, the identification through the WBCs of a PD-linked biochemical profile may benefit the diagnosis and the design of more efficient therapeutic strategies. 

## 4. Materials and Methods

### 4.1. Study Population and Blood Sample Collection

Human whole blood samples were obtained from 42 patients affected by PD, recruited at the Movement Disorders Unit of the Service of Neurology of the University Hospital San Cecilio (Granada, Spain). Inclusion criteria comprised: the diagnosis of PD, according to the UK PD Society Brain Bank [46]; no reported family history of PD; age between 50 and 80 years old; in Hoehn and Yahr (HY) stages I and II (early stages) [47]; and under treatment with dopamine agonists and levodopa. The control group comprised 37 healthy individuals, partners of PD patients, and/or volunteers with no clinical or neurological signs of parkinsonism. They were age- and sex-matched with PD patients. All the individuals under study were negative for *GBA* mutations. The study was approved by the Human Ethics Committee of the hospital and written informed consent was obtained from all participants.

### 4.2. White Blood Cells Extraction

Blood samples were drawn from the cubital vein into EDTA tubes and refrigerated at 4 °C until processed. For white blood cell (WBC) extraction, up to 4 mL of each blood sample were processed in Falcon tubes containing 6 mL of lysis solution for red blood cells (RBC) (RBC lysis solution: 1 mM EDTA solution obtained by a 0.5 M EDTA solution at pH 8.0, prepared with EDTA disodium salt dihydrate from Sigma E-5134, and stored at room temperature). Tubes were mixed by inversion 2 times, incubated for 2 min at room temperature, mixed once more time by inversion and centrifuged at 2000 g (3161 rpm) for 20 min at room temperature. After centrifugation, supernatants were removed, leaving the WBC pellets at the bottom of the tubes. This procedure was repeated at least 3 or 4 times. Cells were suspended in 1 mL of 1 M NaCl Solution, aliquoted and frozen at −20 °C until use.

### 4.3. Enzymatic Assays

After extraction, NaCl Solution was removed from WBC pellets, and then they were resuspended in RIPA lysis buffer (150 Mm NaCl, 1.0% Triton X-100, 0.5% sodium deoxycholate, 0.1% SDS, 50 Mm Tris pH 7.5) together with phosphatase inhibitors (Halt phosphatase inhibitor cocktail, Pierce). Samples were incubated on ice for 30 min and subsequently centrifuged at maximum speed for 20 min at 4 °C. Supernatants were collected and total protein amount was measured using a Bicinchoninic acid (BCA) protein assay kit (Pierce BCA Protein Assay Kit, ref. 23227, Thermo Scientific, Madrid, Spain) according to the manufacturer’s instructions. All samples were tested for the following lysosomal enzymes: β-GCase and β-gal enzymatic activities were measured as previously described [22], whereas CatD and Hex B enzymatic activities were measured by commercial enzymatic assay kits (Cathepsin D Activity Assay Kit ab65302 from Abcam and β-N-Acetylglucosaminidase Assay Kit CS0780 from Sigma-Aldrich). Briefly, 20 µL per sample were incubated in presence of 40 µL of substrate (4-methylumbelliferyl-β-D-glucopyranoside in citrate/phosphate buffer pH 4.5 plus 0.2% taurodeoxicolate for GCase enzymatic activity, and 4-methylumbelliferyl-β-D-galactoside in citrate/phosphate buffer pH 4.5 for β-gal enzymatic activity) in a black 96-well plate for 90 min at 37 °C. Thereafter, reaction was stopped, 240 µL of a stop buffer solution (0.2 M Glycine-NaOH, pH 10.4) was added, and fluorescence was measured on a TECAN infiniteM200PRO plate reader (360 nm excitation; 446 nm emission). Activity was expressed as nanomoles of substrate hydrolyzed per minute per milligram of protein (nmol/min/mg). Regarding the commercial enzymatic assay kits, Hex B activity was measured after the hydrolysis of the substrate N-acetyl-β-D-glucosaminidase, releasing *p*-nitrophenol, which was detected colorimetrically at 405 nm emission. Activity was expressed in units per milliliter (one unit refers to 1.0 µmole of substrate hydrolysed to p-nitrophenol per minute at pH 4.7 and 37 °C). CatD activity was quantified after the cleavage of its synthetic substrate containing the sequence GKPILFFRLK (Dnp)-D-R-NH2, labelled with the fluorescent compound 7-Methoxycoumarin-4-acetic acid (MCA) and using a fluorescence plate reader at 328 nm excitation/460 nm emission. Activity was expressed as relative fluorescence units (RFU) per milligram of protein per millilitre (RFU/mg/mL). All measurements were performed in triplicates. The specific details of every enzymatic assay are summarized in the Table 2. 

### 4.4. Antibodies

The antibodies used were as follows: rabbit anti-human GBA antibody (sc-32883; Santa Cruz Biotechnology), mouse anti-human BGAL antibody (sc-377257; Santa Cruz Biotechnology), mouse anti-human CTSD antibody (sc-377299; Santa Cruz Biotechnology), rabbit anti-human HEXB antibody (PA-36146; ThermoFisher), rabbit anti-human Saposin C antibody (sc-32875; Santa Cruz Biotechnology), mouse anti-human β-Actin antibody (A1978-200UL; Sigma-Aldrich), anti-rabbit antibody (sc-2004; Santa Cruz Biotechnology), and anti-mouse antibody (A2304; Sigma-Aldrich). All antibodies were used at 1:500 in 5% protein blocker in Tris-buffered saline with 0.1 % Tween 20, except for mouse anti-human β-Actin antibody, which was used at 1:50.000 in 5% protein blocker in Phosphate buffered saline with 0.1% Tween 20.

### 4.5. Western Blot Analysis

The protein concentration of WBC lysates was measured by BCA protein assay (Pierce BCA Protein Assay Kit, 23225 ThermoFisher Scientific, Madrid, Spain). An amount of 15 µg of proteins was loaded in every lane of 10% SDS-PAGE gels. After electrophoresis, proteins were transferred to PVDF membranes (0.2 µm Immuno-Blot PVDF Membrane 162-0177 Bio-Rad) for 30 min. Membranes were blocked in 5% non-fat dried milk for 1 h, and then incubated in the appropriate primary antibodies overnight. Immunoblots were performed for proteins GBA, BGAL, HEXB, CTSD, α-Syn, and β-Actin as protein loading controls. Furthermore, PSAP and Saposin C were also assayed due their activator role in the hydrolysis of glucosylceramide and galactocerebroside by β-GCase and β-gal enzymes, respectively. Membranes were developed after incubation with the respective horse radish peroxidase-linked secondary antibodies using enhanced chemiluminescence (ECL). Bands were visualized using the digital imaging system Kodak Image Station 4000 MM PRO and then quantified using the ImageJ software (v. 1.52, NIH, https://imagej.nih.gov/ij/index.html).

### 4.6. Statistical Analysis

Statistical analysis was performed using GraphPad Prism, version 5 (GraphPad Software, San Diego, CA, USA). All quantitative data were averaged across multiple experiments, specifying the number of experiments in the corresponding figure legends. Error bars represent the standard error of the mean. To determine statistically significant differences between groups, *p* values were calculated using the non-parametric Mann–Whitney U test and the one-way ANOVA with Kruskal–Wallis post hoc test. Differences were considered significant at *p* < 0.05. The Spearman correlation coefficient was used when appropriate to test the association between demographic and experimental variables.

## Figures and Tables

**Figure 1 ijms-23-12004-f001:**
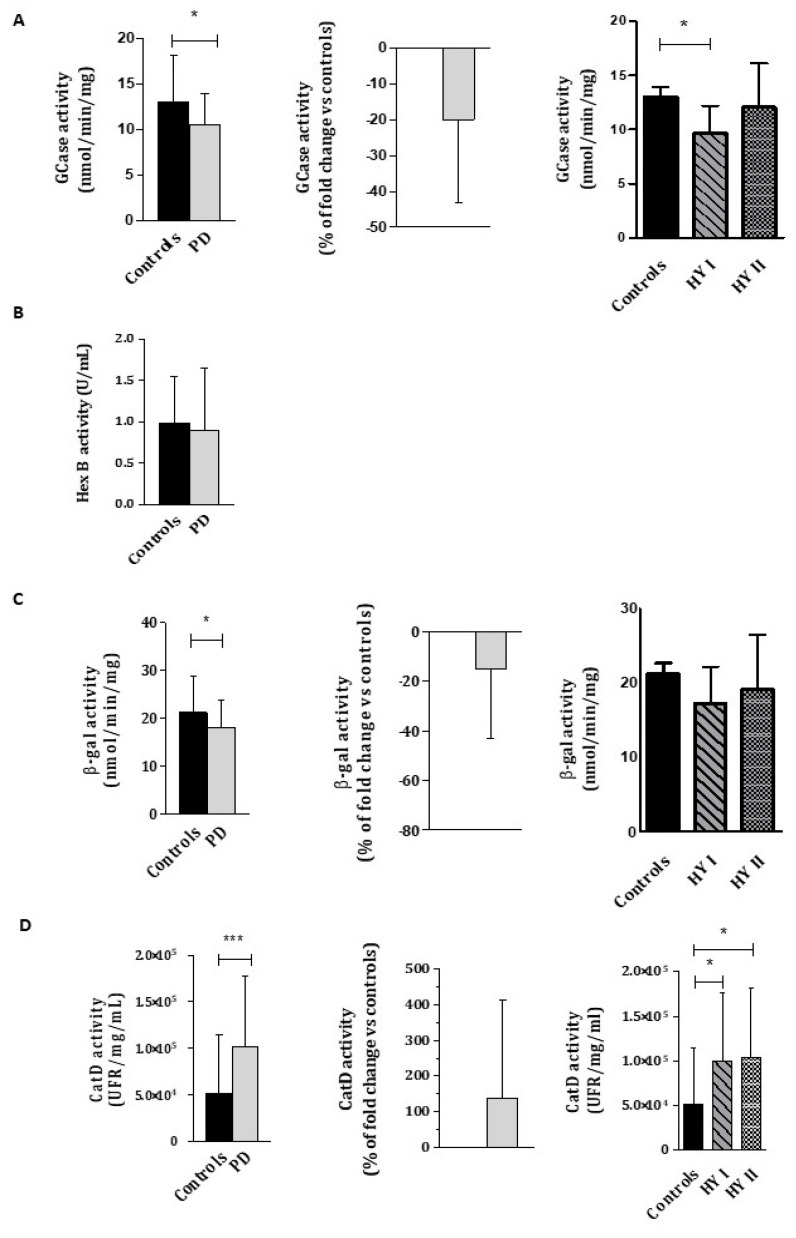
Lysosomal enzymatic activity of β-GCase (**A**), Hex B (**B**), β-gal (**C**), and CatD (**D**) in WBC from PD patients compared to healthy controls. The percentage of fold change over controls was calculated as follows: (PD value—control activity average)/control activity average × 100. Data show mean ± SEM. * *p* < 0.05; *** *p* < 0.001.

**Figure 2 ijms-23-12004-f002:**
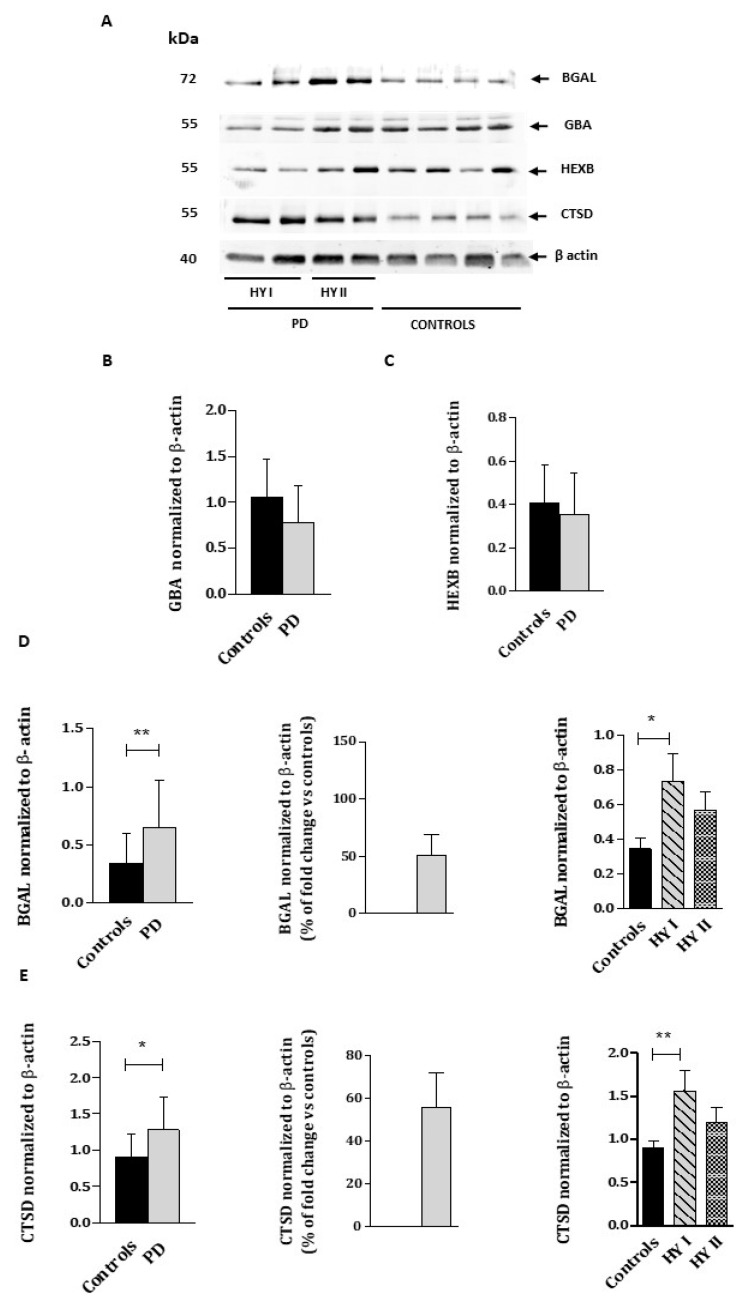
Immunoblot analysis of GBA, BGAL, HEXB, and CTSD proteins (**A**) from the WBC lysates of PD patients and healthy controls. Molecular weight markers are indicated in kilodaltons. Quantification of GBA (**B**), BGAL (**C**), HEXB (**D**), and CTSD (**E**) protein levels normalized to loading control β-actin. HYI and HYII corresponded PD patients at Hoehn and Yahr stages I and II. Quantifications were based on four independent experiments (N = 16 per group). The percentage of fold change over controls was calculated as follows: (PD value– control protein average)/control protein average × 100. Values represent mean ± SEM. * *p* < 0.05, ** *p* < 0.01.

**Figure 3 ijms-23-12004-f003:**
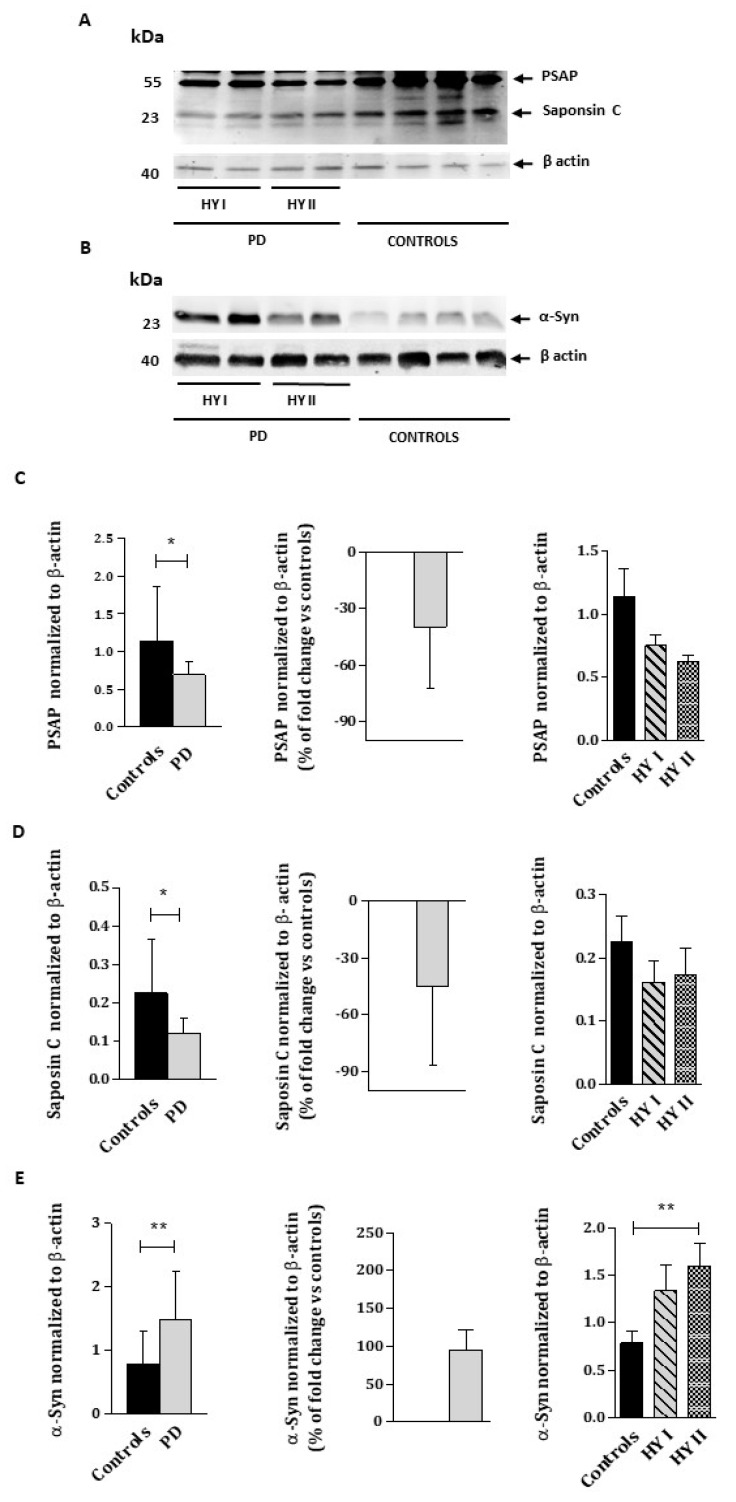
Immunoblot analysis of PSAP, Saposin C (**A**), and α-Syn (**B**) protein levels from the WBC lysates of PD patients and healthy controls. Molecular weight markers are indicated in kilodaltons. The quantification of PSAP (**C**), Saposin C (**D**), and α-Syn (**E**) protein levels normalized to loading control β-actin. HYI and HYII correspond to PD patients at Hoehn and Yahr stages I and II. Quantifications were based on four independent experiments (*n* = 16 per group). The percentage of fold change over controls was calculated as follows: (PD value—control protein average)/control protein average × 100. Values represent mean ± SEM. * *p* < 0.05, ** *p* < 0.01.

**Figure 4 ijms-23-12004-f004:**
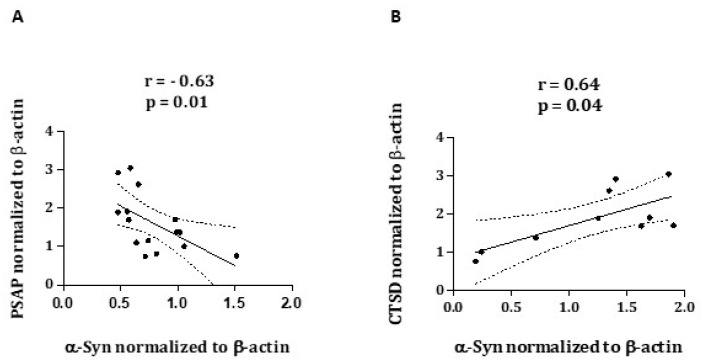
Correlation analysis of PSAP and α-Syn (**A**) and CTSD and α-Syn (**B**), obtained from Western blots from 15 and 10 PD patients, respectively.

**Table 1 ijms-23-12004-t001:** Demographic and clinical data of the study cohort.

	Controls	PD Patients	*p*-Value
Number of individuals	37	42	NA
Men	11	22	0.06
Women	26	20	
* Age (y)	68.3 ± 7.44	66.4 ± 10.28	0.35
* Age at onset (y)	-	60.06 ± 10.77	NA
Hoehn and Yahr stage			NA
I	-	21	
II	-	21	
* WBC total protein µg/µL	0.96 ± 0.33	1.16 ± 0.39	0.01

* Data were reported as mean ± standard deviation; NA: not applicable.

**Table 2 ijms-23-12004-t002:** Specific details of every enzymatic assay.

Lysosomal Enzyme	Substrate	Reaction Buffer	Stop Solution	Sample (µL)	Substrate (µL)	Stop Solution (µL)	Time (min)	Temperature (°C)
β-GCase	3 mM 4-methylumbelliferyl-β-D-glucopyranoside	Citrate/phosphate buffer pH 4.5 plus 0.2% taurodeoxicolate	0.2 M Glycine-NaOH pH 10.4	20	40	240	90	37
β-gal	3 mM 4-methylumbelliferyl-β-D-galactoside	Citrate/phosphate buffer pH 4.5	0.2 M Glycine-NaOH pH 10.4	20	40	240	90	37
Hex B	1 mg/mL N-acetyl-β-D-glucosaminidase	Citrate buffer solution (C2488)	Sodium carbonate solution (S2127)	5	95	200	10	37
CatD	1 mM GKPILFFRLK (Dnp)-D-R-NH2) with MCA	CD Reaction Buffer(ab65302)	-	50	52	-	90	37

## Data Availability

The data from lysosomal activities presented in this study are available on request from the corresponding author.

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
