# Peer review of "Saposin C, Key Regulator in the Alpha-Synuclein Degradation Mediated by Lysosome"

_ijms, 2022, doi:10.3390/ijms231912004_

Round 1
Reviewer 1 Report
Thank you for giving me the opportunity to review the manuscript "Saposin C, key regulator in the alpha-synuclein degradation mediated by lysosome".
The idea of exploring the effect of Saposine C on lysosomal enzyme activity in patients with PD is attractive, however it should be considered that the determinations are being performed in peripheral blood leukocytes and not in brain. The authors should consider these limitations in an additional section.
The main problem of the work is methodological because, being a common disease, it is necessary to calculate the sample size for cases and controls in which the results are conclusive. The disease has an estimated incidence of 300/100.000.
Secondly, and importantly, the inclusion criteria of the patients are not detailed. In the methods section do not refer to the genetic analysis performed to detect the presence of variants in the GBA gene. This is relevant because approximately 30% of patients with PD are carriers of variants in GBA and their GCase activity in these cases is lower than in the control population and is fundamental for the interpretation of the results in this study.
Minor comments: the name of genes must be written in italics, for example GBA gene, PSAP.... needs to be corrected in the entire text.
In page 2 line 59 "GD is a lysosomal storage disorder caused by homozygous 59 mutations" is incorrect, the variants that cause the disease can be the same in the two alleles (homozygous) or different (heterozygous).
Reviewer 2 Report
In the current submission, Ruz et al., present evidence that dysfunctions in lysosomes contribute to the disease pathogenesis in sporadic PD patients. They use white blood cells (WBCs) derived from PD patients and age-matched healthy controls and show that the levels of Saposin C that facilitate interactions between lysosomal enzymes and lipids (majorly sphingolipids) for degradation are decreased in PD patients, affecting beta-glucocerebrosidase (GBA) function and α-synuclein (α-Syn) clearance.
Overall, the topic that the authors have worked on and the conclusion that they have made are interesting. However, my major criticism is on the material/model that they have used in this study – WBCs. Though the results from various assays the authors perform indicate a reduction in lysosomal function and an increase in α-Syn levels, WBCs are not the ideal cell type for such studies as they do not completely recapitulate the scenario observed in the brain cells (neurons). Though it is difficult to obtain tissue (brain) sections from living individuals for the study, the authors should make some efforts and validate their results using patient-derived iPSCs and/or postmortem tissues.
Specific comments:
1. In the result section (Line 94-95) the authors indicate that in PD patients the total WBC protein concentration is higher than in controls. It is not clear how the author normalized the protein concentration from the PD patients with that of the controls while performing various experiments, especially in the assays. For instance, in the materials & methods section the author state that they use 20 µl of sample/lysate as the starting material for the assay (& not the same protein concentration). Can the authors explain this?
2. Figure 1 – Legends and text descriptions are missing for the 3rd graph in panels A, C, and D.
3. Figure 1 & Figure 3 – Can the authors explain how the % fold change values (Figure 1, Panel A, C, D – 2nd graph; Figure 3 – Panel C, D E – 2nd graph) were calculated in the materials and methods section?
4. Figure 2 – Panel A: An antibody against CathepsinD (CTSD), detects two bands. A immature band at around 45-50 kDa represents the pro-/intermediate form while the 34 kDa heavy chain band represents the functional, mature CTSD. However, this band of mature CTSD is missing. It is possible that in the controls there is more mature CTSD while due to some trafficking defects in PD patients the immature CTSD is higher while the levels of mature CTSD is lower in the lysosomes. It is therefore extremely important that the authors quantify, present the mature bands and make conclusions based on those results.
5. Line 187-188: Can the author provide some explanation on why there is an increase in BGAL and CTSD protein levels in HY stage I and not is HY stage II when compared to the controls.
6. Line 210-211: “However, when this activity is explored in periphery, results are incongruous” Can the authors clarify/better explain the above statement? It is not clear what the author mean/refer to as “periphery”.
7. Line 269-271: Can the authors clarify on how they came to this conclusion? It is not clear how the increase in monohexosulceramide levels was measured.
8. Line 324-325: From the description of the assay and table 2 it appears that the authors measured fluorescence levels only after 90 min. However, the activity of the enzyme (B-Gal) is expressed as per minute per milligram of protein. Can the authors clarify it?
Minor comments:
Line 42-43: Check (requires rephrasing)
Expansion for GBA missing
Line 60: ….mutations in the same GBA gene.
Line 217-219: Check (requires rephrasing)
Line 237-240: Check (requires rephrasing)
Line 249: ….the lysosome where IT is processed….
Line 274-275: It is remarkable that….
Section 4.3. Enzymatic assays: The degree sign (°) has to be superscripted in this entire section
Round 2
Reviewer 2 Report
In this revised version Ruz et al., have attempted to address some of the comments that where raised. However, their response to my major comment (point 1 – response to reviewer 2) is not convincing/satisfactory. I therefore leave the decision to the editor with respect to the next step. I still stand with my comment that the major finding presented in this manuscript (a reduction in lysosomal function and an increase in α-Syn levels) has to be validated at least on postmortem tissues (controls, patients).
